# Colchicine for the treatment of patients with COVID-19: an updated systematic review and meta-analysis of randomised controlled trials

Huzaifa Ahmad Cheema [ORCID],[1] Uzair Jafar,[1] Abia Shahid,[1] Waniyah Masood,[2] Muhammad Usman,[1] Alaa Hamza Hermis,[3] Muhammad Arsal Naseem,[1] Syeda Sahra,[4] Ranjit Sah,[5] Ka Yiu Lee [ORCID] [6]

HAC and UJ contributed equally.

HAC and UJ are joint first authors.

For numbered affiliations see end of article.

**Correspondence to**
Dr Ka Yiu Lee; kyle.lee@miun.se

## ABSTRACT

**Objectives** We conducted an updated systematic review and meta-analysis to investigate the effect of colchicine treatment on clinical outcomes in patients with COVID-19.

**Design** Systematic review and meta-analysis.

**Data sources** We searched PubMed, Embase, the Cochrane Library, medRxiv and ClinicalTrials.gov from inception to January 2023.

**Eligibility criteria** All randomised controlled trials (RCTs) that investigated the efficacy of colchicine treatment in patients with COVID-19 as compared with placebo or standard of care were included. There were no language restrictions. Studies that used colchicine prophylactically were excluded.

**Data extraction and synthesis** We extracted all information relating to the study characteristics, such as author names, location, study population, details of intervention and comparator groups, and our outcomes of interest. We conducted our meta-analysis by using RevMan V.5.4 with risk ratio (RR) and mean difference as the effect measures.

**Results** We included 23 RCTs (28 249 participants) in this systematic review. Colchicine did not decrease the risk of mortality (RR 0.99; 95% CI 0.93 to 1.05; $I^2$=0%; 20 RCTs, 25 824 participants), with the results being consistent among both hospitalised and non-hospitalised patients. There were no significant differences between the colchicine and control groups in other relevant clinical outcomes, including the incidence of mechanical ventilation (RR 0.75; 95% CI 0.48 to 1.18; p=0.22; $I^2$=40%; 8 RCTs, 13 262 participants), intensive care unit admission (RR 0.77; 95% CI 0.49 to 1.22; p=0.27; $I^2$=0%; 6 RCTs, 961 participants) and hospital admission (RR 0.74; 95% CI 0.48 to 1.16; p=0.19; $I^2$=70%; 3 RCTs, 8572 participants).

**Conclusions** The results of this meta-analysis do not support the use of colchicine as a treatment for reducing the risk of mortality or improving other relevant clinical outcomes in patients with COVID-19. However, RCTs investigating early treatment with colchicine (within 5 days of symptom onset or in patients with early-stage disease) are needed to fully elucidate the potential benefits of colchicine in this patient population.

**PROSPERO registration number** CRD42022369850.

### STRENGTHS AND LIMITATIONS OF THIS STUDY

⇒ This systematic review used a comprehensive search for articles evaluating colchicine treatment for patients with COVID-19.

⇒ We performed, to our knowledge, the largest meta-analysis to date by pooling results from 23 trials.

⇒ There was significant interstudy heterogeneity in the overall analysis of some outcomes, such as length of hospitalisation.

⇒ Our results may not be generalisable to the use of colchicine early in the course of the disease as most randomised controlled trials studied late-stage treatment.

## INTRODUCTION

Unprecedented global research has been conducted as a result of the COVID-19 pandemic. Even though particular antiviral medications, immunomodulators and monoclonal antibodies have been approved to treat COVID-19,[1–4] ongoing clinical trials are being conducted to combat the virus more successfully, particularly using repurposed drugs and herbal medicines that provide a cheap therapeutic alternative.[5–8] Limited knowledge of COVID-19 molecular mechanisms and pathophysiology, lack of definitive supporting evidence and expensive costs of some treatment options are some of the challenges faced in using different drugs for COVID-19.[9 10]

Colchicine, an anti-inflammatory medication commonly used to treat gout, recurrent pericarditis and familial Mediterranean fever, is being extensively investigated among COVID-19 patients.[11] Inhibiting neutrophil chemotaxis, suppressing inflammasome signalling and preventing the cytokine storm through microtubule depolarisation are some of the potential benefits of colchicine in treating COVID-19.[11] Currently, guidelines

from the WHO recommend against treatment with colchicine based on data from 10 studies.[1]

Although colchicine has been the subject of multiple clinical trials in COVID-19 patients, findings from smaller randomised controlled trials (RCTs) that were positive were later contradicted by results from major RCTs.[11] Therefore, we performed this meta-analysis to evaluate the safety and efficacy of colchicine for the treatment of COVID-19 patients and to interpret the significance of the latest data by incorporating it into our review.

## METHODS

Our systematic review and meta-analysis was registered with PROSPERO (CRD42022369850) and conducted by following the recommendations of Preferred Reporting Items for Systematic Reviews and Meta-Analyses (PRISMA).[12] The PRISMA checklist is provided in online supplemental table 1.

### Data sources and search strategy

We searched MEDLINE (via PubMed), Embase, the Cochrane Library, medRxiv and ClinicalTrials.gov from inception to January 2023 using a search strategy consisting of terms related to "colchicine" and "COVID-19" without any restrictions. Grey literature and the reference lists of the relevant studies were also explored to include all eligible trials. We did not restrict our search based on language. The detailed search strategy for different databases is given in online supplemental table 2.

### Eligibility criteria

The inclusion criteria were (1) population: patients with a diagnosis of COVID-19; (2) intervention: colchicine regardless of dosing or regimen; (3) comparator: placebo or standard of care; (4) outcome: reporting any outcome of interest and (5) study design: RCTs only. Observational studies, quasi-randomised trials and studies that used colchicine prophylactically were excluded.

### Study selection

All the literature obtained from our searches was imported into Mendeley Desktop V.1.19.8 and duplicates were removed. The remaining articles were subjected to a rigorous screening process by two independent reviewers in a two-stage process (title/abstract screening followed by full-text screening).

### Data extraction and outcomes

We extracted all information relating to the study characteristics such as author names, location, study population, details of intervention and comparator groups, and our outcomes of interest. We chose all-cause mortality as our primary outcome while the secondary outcomes included the incidence of mechanical ventilation (MV), risk of intensive care unit (ICU) admission, risk of hospitalisation, the length of hospital stay and the rate of no

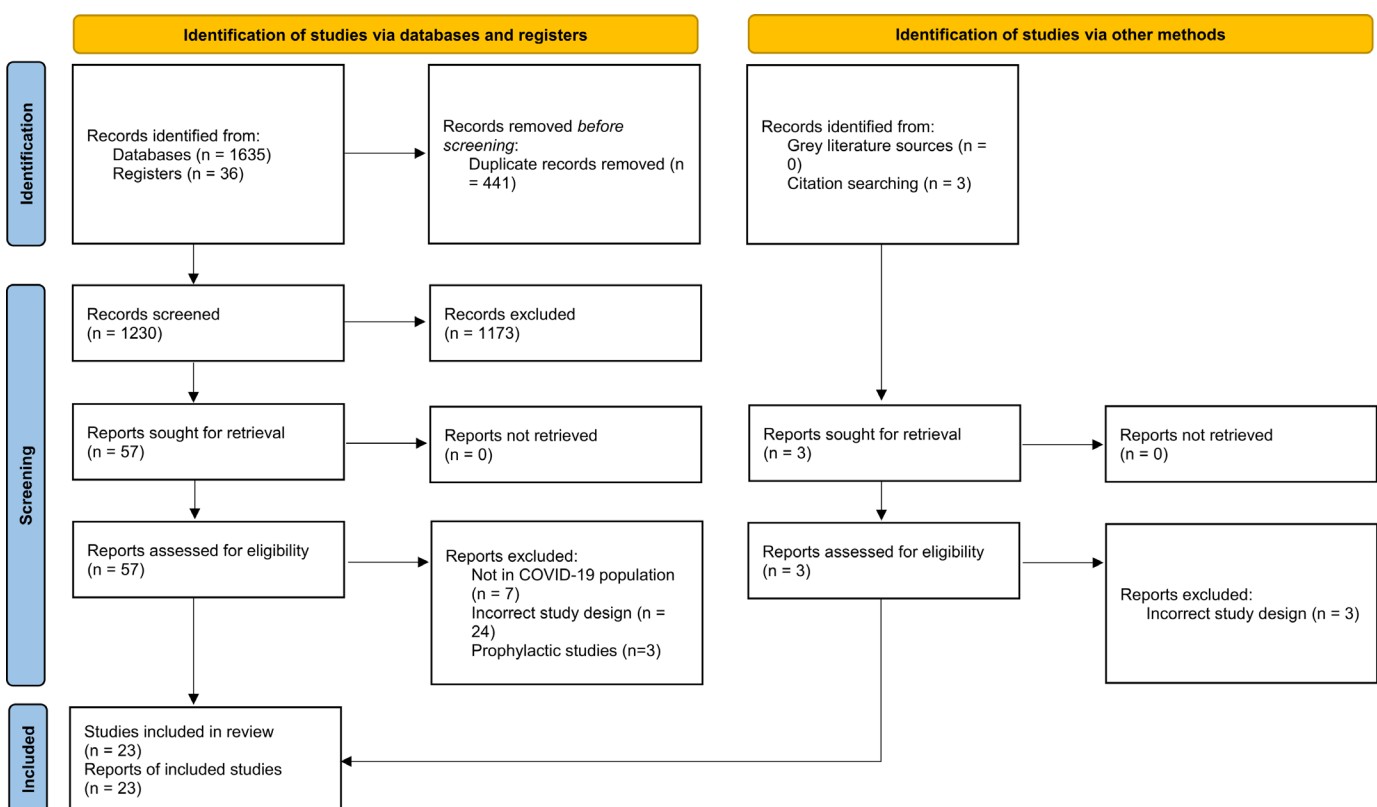

**Figure 1** PRISMA 2020 flow chart. PRISMA, Preferred Reporting Items for Systematic Reviews and Meta-Analyses.

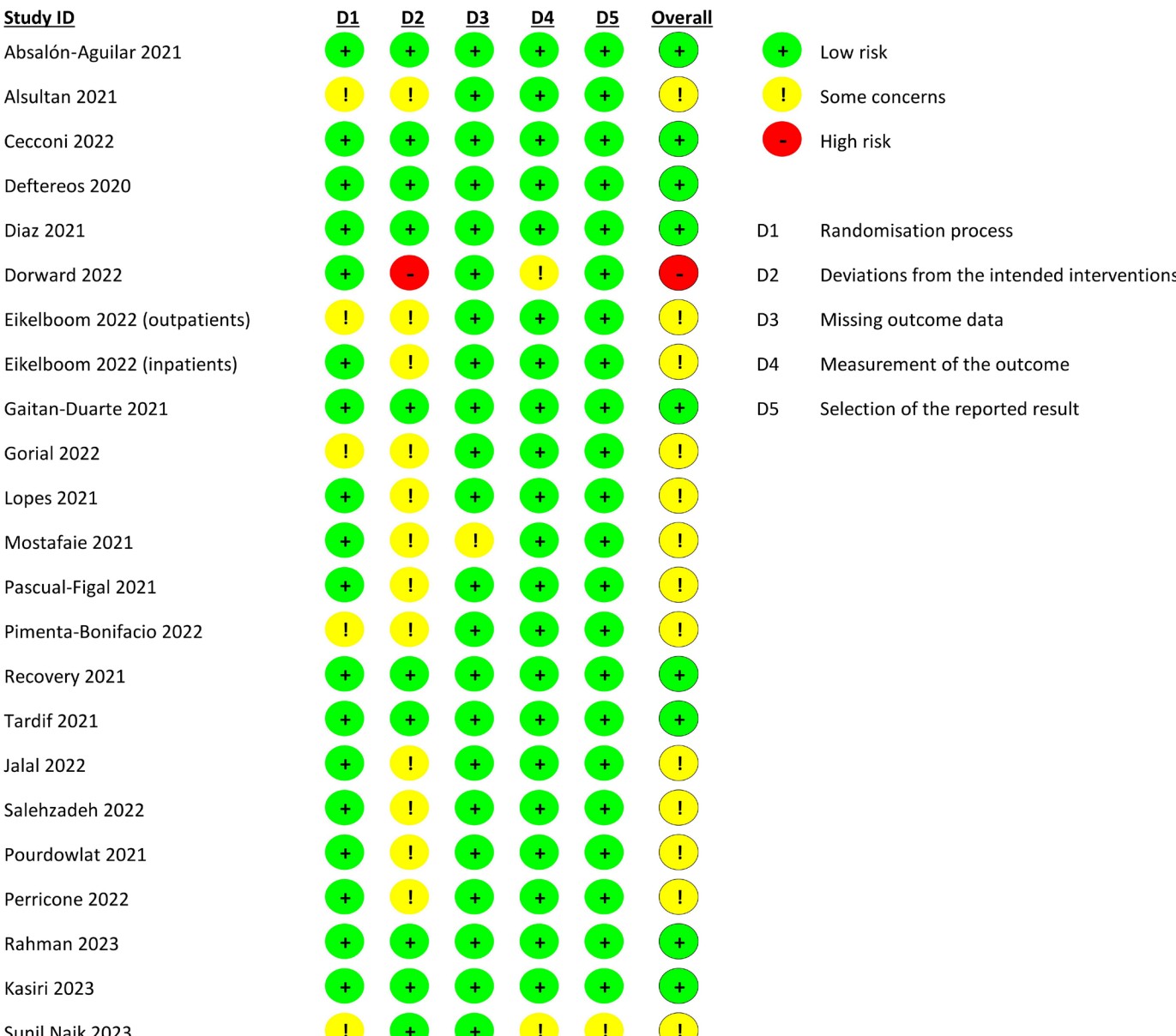

**Figure 2** Quality assessment of included trials.

recovery (proportion of patients with no symptomatic resolution at the end of study).

## Quality assessment

The risk of bias was assessed using the revised Cochrane Risk of Bias tool (RoB 2.0).[13] A rating of low risk of bias, some concerns or a high risk of bias was assigned to each study using five domains: randomisation process, deviations from intended interventions, missing outcome data, measurement of the outcome and selection of reported result.

## Data analysis

We conducted our meta-analysis using RevMan V.5.4 with risk ratio (RR) and mean difference (MD) as the effect measures for categorical and continuous variables, respectively. We used a random-effects model as we anticipated our included studies to be substantially heterogeneous. We evaluated heterogeneity using the $\chi^2$ test, the $I^2$ statistic and the $tau^2$ estimate. We used the Cochrane Handbook for Systematic Reviews of Intervention to interpret the values of the $I^2$ statistic.[14] We conducted a subgroup analysis for our primary outcome on the basis of the study population (hospitalised vs non-hospitalised patients). In addition, we conducted a sensitivity analysis on our primary outcome by excluding trials that used colchicine in combination with another intervention. For outcomes with data from 10 studies or more, we assessed publication bias by constructing funnel plots and running Egger's test for funnel plot asymmetry.

## Patient and public involvement

None.

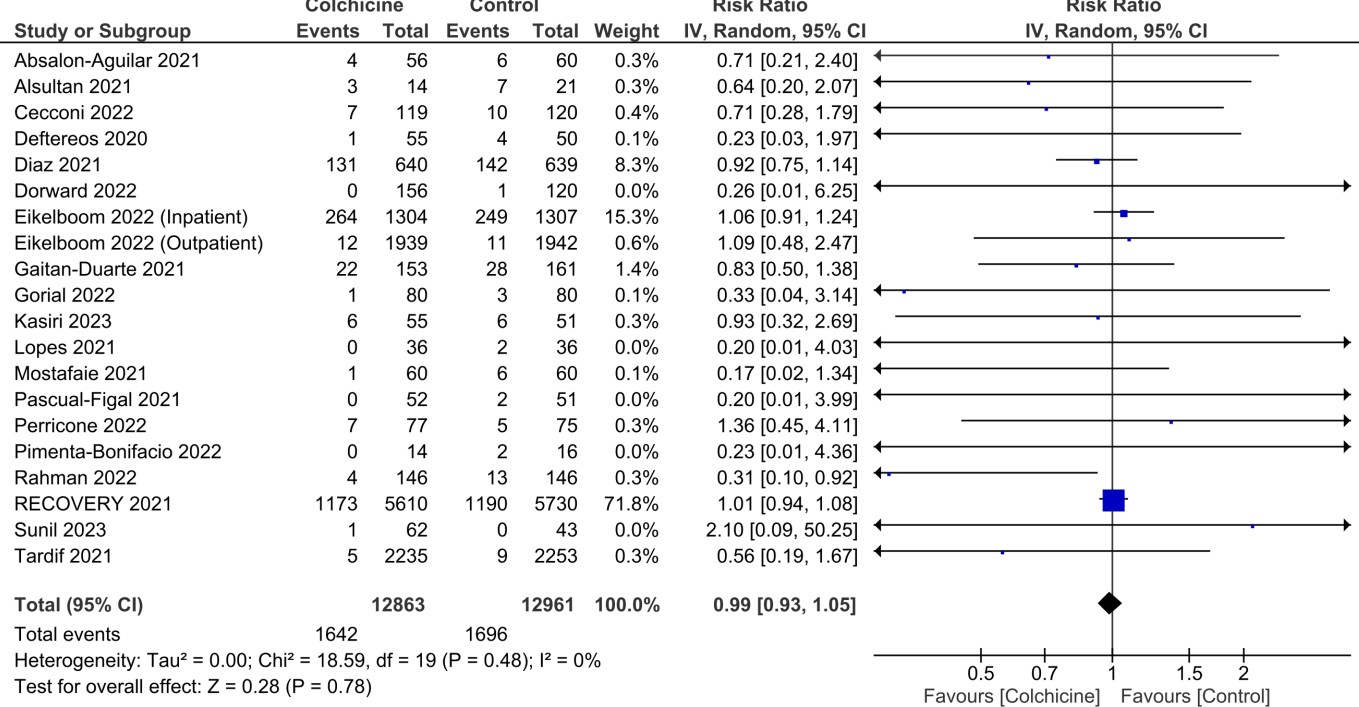

**Figure 3** Effect of colchicine on all-cause mortality in patients with COVID-19.

# RESULTS

## Search results and study characteristics

A total of 23 RCTs (28 249 participants) were eligible for inclusion in our meta-analysis.[15–37] The details of the screening process are presented in figure 1.

17 RCTs were conducted in hospitalised patients, 4 evaluated outpatients only[17 21 22 30] and 1 RCT included both hospitalised and non-hospitalised patients.[26] Most of the trials had small sample sizes while the RECOVERY trial was the largest RCT with 11 340 patients.[23] The trials employed a variety of colchicine regimens and most were open-label that used standard of care as the comparator. Two RCTs combined colchicine with other treatments; one with rosuvastatin[25] and the other with herbal phenolic monoterpene.[33] The characteristics of each RCT are summarised in online supplemental table 3.

## Risk of bias assessment

According to RoB 2.0, 7 studies were at a low risk of bias, 1 had a high risk of bias and 12 had some concerns about bias (figure 2). The most common source of bias was lack of blinding and deviations from protocol while some trials also had issues in the randomisation process.

## Results of the meta-analysis

### Primary outcome

Our analysis, considering 25 824 patients from 20 RCTs, failed to find a statistically significant reduction in the risk of mortality with colchicine use (RR 0.99; 95% CI 0.93 to 1.05; p=0.78; $I^2$=0%; figure 3). The Egger's test indicated that publication bias was present (p=0.002; online supplemental figure 1). The results were consistent ($P_{interaction}$=0.59) among both hospitalised (RR 0.98; 95% CI 0.90 to 1.06; p=0.64; $I^2$=5%) and non-hospitalised patients (RR 0.82; 95% CI 0.43 to 1.55; p=0.54; $I^2$=0%; online supplemental figure 2).

Sensitivity analysis by excluding two studies that used combination therapies in the intervention group,[25 33] produced results consistent with our main analysis (RR 1.00; 95% CI 0.94 to 1.06; p=0.88; $I^2$=0%). The results remained congruous with our primary analysis after performing a second sensitivity analysis excluding studies with a high risk of bias or some concerns about bias in the domain of deviations from intended interventions (RR 0.97; 95% CI 0.88 to 1.06; p=0.46; $I^2$=3%).

### Secondary outcomes

There were no significant differences between the colchicine and control groups in the incidence of MV (RR 0.75; 95% CI 0.48 to 1.18; p=0.22; $I^2$=40%; 8 RCTs, 13 262 participants; online supplemental figure 3), risk of ICU admission (RR 0.77; 95% CI 0.49 to 1.22; p=0.27; $I^2$=0%; 6 RCTs, 961 participants; online supplemental figure 3), risk of hospital admission (RR 0.74; 95% CI 0.48 to 1.16; p=0.19; $I^2$=70%; 3 RCTs, 8572 participants; online supplemental figure 4), length of hospital stay (MD −0.70 days; 95% CI −2.30 to 0.89 days; p=0.39; $I^2$=90%; 8 RCTs, 1028 participants; online supplemental figure 5) and the rate of no recovery (RR 0.77; 95% CI 0.56 to 1.05; p=0.10; $I^2$=77%; 5 RCTs, 13 109 participants; online supplemental figure 6).

# DISCUSSION

To the best of our knowledge, this is the largest and most up-to-date meta-analysis to assess the effectiveness of colchicine for COVID-19. The results of this meta-analysis

suggest that colchicine does not have a significant effect on the risk of mortality in patients with COVID-19. There was also no significant benefit of colchicine in reducing the risk of MV, ICU admission, hospital admission or improving recovery rates.

Colchicine's regulation of the inflammatory response has been proposed as a means to inhibit the inflammation caused by COVID-19. The primary mechanism by which colchicine affects cells is through the inhibition of microtubule polymerisation, which has a ripple effect on various cellular functions such as maintaining cell shape, signalling, cell division, migration and transport.[11 38] Additionally, colchicine modulates the synthesis of inflammatory cytokines such as tumour necrosis factor (TNF), interleukin-1 (IL-1) and interleukin-6 (IL-6) and disrupts the activation of inflammasomes, neutrophil adhesion and recruitment.[11 38] Given that COVID-19 is characterised by excessive inflammation and a cytokine response, colchicine's anti-inflammatory effects may have the potential to improve outcomes.

Our results are consistent with some of the previous meta-analyses which also demonstrated no significant clinical benefit of colchicine in COVID-19 patients.[39–43] On the other hand, the meta-analyses by Zein and Raffaello and Golpour et al[44 45] found that colchicine use was associated with a reduction in the risk of mortality, and Kow et al demonstrated a shorter duration of hospital stay with colchicine.[46] However, the meta-analysis by Zein and Raffaello and Golpour et al also incorporated observational studies, which increases the likelihood of confounding bias in their findings.[44 45] Additionally, all of the previous meta-analyses included a small number of RCTs with most including less than or equal to 11.[47] Conversely, our meta-analysis pooled results from 23 RCTs, therefore, including a significantly greater cumulative sample size and substantially increasing the confidence in our estimates.

A recent umbrella review found that exposure to colchicine was associated with reduced mortality.[47] However, this discrepancy can be explained by the fact that it pooled estimates from meta-analyses instead of primary studies. Since there is a significant overlap in the included studies between the prior meta-analyses, pooling meta-analysed effect sizes leads to erroneous and flawed results due to double-counting of outcome data. The correct approach would be to synthesise data from the primary studies to provide valid estimates.[48]

It is worth noting that despite including a large number of RCTs, the results of our meta-analysis are still greatly influenced by the RECOVERY trial.[23] The RECOVERY trial is by far the largest trial of colchicine which found no clinical benefit of colchicine in COVID-19 patients. However, this was a very late-stage treatment RCT with the median time from symptom onset to treatment initiation being 9 days. It is well recognised that late initiation of therapy might result in decreased effectiveness of antivirals.[49] Therefore, early treatment (within 5 days of symptom onset or at an early stage of the disease) with colchicine might prove to be beneficial for COVID-19 patients; however, this needs to be investigated in future RCTs as all of the currently available RCTs evaluated late-stage treatment.

Some limitations of our meta-analysis should be noted. The quality of the RCTs included in the meta-analysis was mixed, with 9 being of high quality, 1 being of low quality and 13 having some concerns about bias. This may have affected the reliability of the results. Additionally, there was significant interstudy heterogeneity in the overall analysis of some outcomes, such as length of hospitalisation. Our results may not be generalisable to the usage of colchicine early in the course of the disease as most RCTs are of late-stage treatment; consequently, there was not enough data to compare early-stage versus late-stage treatment. Furthermore, the composite endpoint of death or MV is an important outcome; however, due to a lack of data provided by the RCTs, we were unable to analyse this endpoint. Moreover, many studies either did not report the vaccination status of their participants or included unvaccinated patients only. Therefore, the COVID-19 vaccination status as a potential effect modifier needs to be studied further. Finally, there was evidence of the presence of publication bias in our meta-analysis implying that small studies that did not report positive findings may not have been published.

## CONCLUSION

The results of this meta-analysis do not support the use of colchicine as a treatment for reducing the risk of mortality or improving other clinical outcomes in patients with COVID-19. However, RCTs investigating early treatment with colchicine (within 5 days of symptom onset or in patients with early-stage disease) are needed to fully elucidate the potential benefits of colchicine in this patient population.

**Author affiliations**
[1]Department of Medicine, King Edward Medical University, Lahore, Pakistan
[2]Department of Medicine, Dow University of Health Sciences, Karachi, Pakistan
[3]Nursing College, Al-Mustaqbal University, 51001 Hillah, Babylon, Iraq
[4]Department of Infectious Diseases, The University of Oklahoma Health Sciences Center, Oklahoma City, Oklahoma, USA
[5]Department of Microbiology, Dr. D. Y. Patil Medical College, Hospital and Research Centre, Dr. D. Y. Patil Vidyapeeth, Pune 411018, Maharashtra, India
[6]Swedish Winter Sports Research Centre, Department of Health Sciences, Mid Sweden University, Sundsvall, Sweden

**Contributors** Conception and design of study: HAC, AS, KYL and UJ; acquisition of data: MU, MAN and HAC; data analysis and/or interpretation: SS, RS, UJ, AHH and WM; drafting or writing of the manuscript: MAN, UJ, SS, MU, AS and WM; substantial revision or critical review of the manuscript: HAC, AHH, RS and KYL; guarantor: HAC. All authors have approved the final version of the manuscript.

**Funding** The authors have not declared a specific grant for this research from any funding agency in the public, commercial or not-for-profit sectors.

**Competing interests** None declared.

**Patient and public involvement** Patients and/or the public were not involved in the design, or conduct, or reporting, or dissemination plans of this research.

**Patient consent for publication** Not applicable.

**Provenance and peer review** Not commissioned; externally peer reviewed.

**Data availability statement** Data are available on reasonable request.

**ORCID iDs**
Huzaifa Ahmad Cheema http://orcid.org/0000-0001-6783-7137
Ka Yiu Lee http://orcid.org/0000-0001-5577-0940

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
