## [Reviewer comments · BMJ Open]

ARTICLE DETAILS

TITLE (PROVISIONAL)	Colchicine for the treatment of patients with COVID-19: an updated systematic review and meta-analysis of randomized controlled trials
AUTHORS	Cheema, Huzaifa Ahmad; Jafar, Uzair; Shahid, Abia; Masood, Waniyah; Usman, Muhammad; Hermis, Alaa Hamza; Naseem, Muhammad Ahsan; Sahra, Syeda; Sah, Ranjit; Lee, Ka Yiu

VERSION 1 – REVIEW

REVIEWER	Belletti, Alessandro IRCCS San Raffaele Scientific Institute, Department of Anaesthesia and Intensive Care
REVIEW RETURNED	14-Jul-2023

GENERAL COMMENTS	In this meta-analysis of RCTs, Lee and colleagues examined the effect of colchicine treatment on major outcomes in COVID-19 patients. They found that colchicine did not improve survival, nor had any effect on any of the secondary outcomes. The meta-analysis is well conducted from a methodological point of view, and has been registered on PROSPERO. Submitted review is consistent with PROSPERO registration I have only few comments for the Authors which I hope will help to improve the manuscript: - it would be interesting to provide a sensitivity analysis including only low-risk of bias studies, or blinded studies only- the Authors commented on timing of colchicine treatment. I understand that no available RCT started treatment <5 days after symptom onset. However, it would be interesting to abstract this information and provide it in supplementary table 1- studies on COVID-19 frequently report data on a composite endpoint of mortality or need for mechanical ventilation/ICU admission. Is such analysis feasible with available data?
---

REVIEWER	CORTINA, ANDRE Irmandade da Santa Casa de Misericórdia de Santos, Serviço de Urgência e Emergência
REVIEW RETURNED	14-Jul-2023

GENERAL COMMENTS	What it shows me is the efficiency in controlling the inflammatory cascade caused by COVID-19, which is the worst manifestation carried out by COVID-19 due to all the other effects and problems, having an effectiveness with colchicine in controlling the inflammatory regulation, having as example and we can signal and Chloroquine performing and working in the same aspect. It is
---

	important to point out that the Yellow Fever Vaccine here in Brazil acts in the same way as it blocks the inflammatory cascade when yellow fever is inoculated by its vector. It is worth mentioning that this is not a cure or protection, but a powerful weapon to prevent the inflammatory process from continuing and resulting in earlier hospital discharges for these patients with COVID-19. The start of use, I believe it would not be ideal in view of the onset of the inflammatory cascade, it can be performed between 3 and 5 days from the onset of symptoms, these effects seen in the countless patients treated here by our institution.
--	--

REVIEWER	Neupane, Astha South Dakota State University, Biology and Microbiology
REVIEW RETURNED	10-Oct-2023

GENERAL COMMENTS	Introduction: Needs some more background information Some details need to be included like:  • Challenge of using different drugs during COVID treatment • Why was Colchicine initially used for Covid treatment? • Detail the Mechanism of action of Colchicine, and how it can work on COVID-19? • Safety profile of drugs • Guideline from Regulatory body regarding the use of Colchicine in COVID-19? Methods  • In Prism: Detail instruction for inclusion and exclusion criteria • Why grey literature was used? Search strategies for medRxiv, ClinicalTrials.gov and Grey literature • Why/ how rating of low risk of bias, some concerns or a high risk of bias was assigned to each study using five domains: randomization process, deviations from intended interventions, missing outcome data, measurement of the outcome, and selection of reported result.
--

REVIEWER	Kurth, Tobias Charité Universitätsmedizin Berlin, Institute of Public Health
REVIEW RETURNED	29-Dec-2023

GENERAL COMMENTS	While this is not the first meta-analysis on this topic, this meta-analysis included more randomized controlled trials than previous studies and therefore adds to the current knowledge. The overall quality assessment of the included studies showed that only 9 were of high quality, and most data came from the RECOVERY trial. The study has been registered in PROSPERO. I could not verify this protocol, as PROSPERO was down on the day of review. The study is well conducted overall and followed the PRISMA guidelines. Comments. 1. As the authors discuss, one of the limitations of the study is that many of the included randomized controlled trials are very small. Therefore, pooling across these studies remains problematic and leaves uncertainties about the findings.
--

	2. The authors report that there was significant interstudy heterogeneity and seem to base this interpretation on the I-square statistic. I-square indicates what proportion of the observed variance reflects variance in true effects rather than sampling error. The interstudy heterogeneity (between study variance) is measured with Tau-square. 3. The authors state that their study is the largest meta-analysis to date. I suggest omitting this statement as the number of included studies may not indicate the overall quality, as the authors discuss.
--	--

VERSION 1 – AUTHOR RESPONSE

Responses to the comments of Reviewer #1

1. - it would be interesting to provide a sensitivity analysis including only low-risk of bias studies, or blinded studies only

Response: The authors thank the honorable reviewer for their comments and appreciation of our meta-analysis. We have added a sensitivity analysis for our primary outcome by excluding high risk of bias or unblinded studies noting that the results did not change.

2. - the Authors commented on timing of colchicine treatment. I understand that no available RCT started treatment <5 days after symptom onset. However, it would be interesting to abstract this information and provide it in supplementary table 1

Response: We thank the reviewer for their suggestion. However, due to a lack of adequate reporting, we are unable to abstract detailed information on this.

3. - studies on COVID-19 frequently report data on a composite endpoint of mortality or need for mechanical ventilation/ICU admission. Is such analysis feasible with available data?

Response: The authors thank the honorable reviewer for their comments. Unfortunately, due to variable reporting of outcomes, such analyses of composite outcomes were not feasible.

Responses to the comments of Reviewer #2

1. What it shows me is the efficiency in controlling the inflammatory cascade caused by COVID-19, which is the worst manifestation carried out by COVID-19 due to all the other effects and problems, having an effectiveness with colchicine in controlling the inflammatory regulation, having as example and we can signal and Chloroquine performing and working in the same aspect. It is important to point out that the Yellow Fever Vaccine here in Brazil acts in the same way as it blocks the inflammatory cascade when yellow fever is inoculated by its vector.

It is worth mentioning that this is not a cure or protection, but a powerful weapon to prevent the inflammatory process from continuing and resulting in earlier hospital discharges for these patients with COVID-19.

The start of use, I believe it would not be ideal in view of the onset of the inflammatory cascade, it can be performed between 3 and 5 days from the onset of symptoms, these effects seen in the countless patients treated here by our institution.

Response: We thank the honorable reviewer for commenting on our study. We agree that colchicine might be useful in controlling inflammation cascade of COVID-19 theoretically but it is not supported by our results currently.

Responses to the comments of Reviewer #3

1. Introduction:

Needs some more background information

Some details need to be included like:

- Challenge of using different drugs during COVID treatment
- Why was Colchicine initially used for Covid treatment?
- Detail the Mechanism of action of Colchicine, and how it can work on COVID-19?
- Safety profile of drugs
- Guideline from Regulatory body regarding the use of Colchicine in COVID-19?

Response: We thank the respected reviewer for their valuable suggestions. We have updated our introduction by adding more background detail addressing the above points.

2. Methods

- In Prism: Detail instruction for inclusion and exclusion criteria
- Why grey literature was used? Search strategies for medRxiv, ClinicalTrials.gov and Grey literature
- Why/ how rating of low risk of bias, some concerns or a high risk of bias was assigned to each study using five domains: randomization process, deviations from intended interventions, missing outcome data, measurement of the outcome, and selection of reported result.

Response: We thank the reviewer for their comments. We had already detailed inclusion and exclusion criteria in our methods. However, now we have restructured it and improved it according to the PICOS criteria.

-Grey literature searching is recommended by the Cochrane guidelines. That is why we undertook it. Moreover, search strategies for the aforementioned databases are not required as per the Cochrane and PRISMA guidelines as these are not bibliographic databases and do not have specialized searching tools like the major bibliographic databases. For those (PubMed, Embase and the Cochrane Library), we have already provided detailed search strategies in our Supplementary File.

-The risk of bias assessment was carried out according to the Cochrane guidelines. The details can be found in their explanation paper (Sterne JAC, Savović J, Page MJ, Elbers RG, Blencowe NS, Boutron I, et al. RoB 2: a revised tool for assessing risk of bias in randomised trials. *BMJ* 2019;366:l4898.)

Responses to the comments of Reviewer #4

1. 1. As the authors discuss, one of the limitations of the study is that many of the included randomized controlled trials are very small. Therefore, pooling across these studies remains problematic and leaves uncertainties about the findings.

Response: We thank the respected reviewer for their comments and appreciation of our meta-analysis. We agree that such limitations of the included studies exist. However, the purpose of the meta-analysis is to pool together such small studies which are underpowered on their own, to

generate more reliable and higher-quality findings. Therefore, the value of this meta-analysis cannot be discounted despite the inherent limitations of the studies.

2. 2. The authors report that there was significant interstudy heterogeneity and seem to base this interpretation on the I-square statistic. I-square indicates what proportion of the observed variance reflects variance in true effects rather than sampling error. The interstudy heterogeneity (between study variance) is measured with Tau-square.

Response: We thank the reviewer for their feedback. While we concur that I-square statistic indeed indicates that technically, however, it is still used as a measure of heterogeneity. The Cochrane Handbook states the following interpretation for I² “A rough guide to interpretation in the context of meta-analyses of randomized trials is as follows:

(<https://training.cochrane.org/handbook/current/chapter-10>)

0% to 40%: might not be important;

30% to 60%: may represent moderate heterogeneity*;

50% to 90%: may represent substantial heterogeneity*;

75% to 100%: considerable heterogeneity*.”

Nevertheless, we agree that Tau-square is a better measure of heterogeneity and we have also used it to interpret our results. We have mentioned this in the methods section now as well.

3. 3. The authors state that their study is the largest meta-analysis to date. I suggest omitting this statement as the number of included studies may not indicate the overall quality, as the authors discuss.

Response: As suggested by the respected reviewer, we have omitted the word “largest” from our discussion.

VERSION 2 – REVIEW

REVIEWER	Belletti, Alessandro IRCCS San Raffaele Scientific Institute, Department of Anaesthesia and Intensive Care
REVIEW RETURNED	18-Feb-2024

GENERAL COMMENTS	The Authors now present a revised version of the manuscript. My comments have been adequately addressed. I only have one minor comment: please acknowledge in the discussion that unfortunately there were not enough data to analyse early vs late treatment, or to analyse the composite endpoint of death or mechanical ventilation.
--

VERSION 2 – AUTHOR RESPONSE

Responses to the comments of Reviewer #1

1. My comments have been adequately addressed. I only have one minor comment: please acknowledge in the discussion that unfortunately there were not enough data to analyse early vs late treatment, or to analyse the composite endpoint of death or mechanical ventilation.

Response: The authors thank the honorable reviewer for their additional comments and appreciation of our revision. With this we have further improved our paper. We have now acknowledged the above-mentioned limitations in our discussion section.